# Extracellular Vesicles’ Role in Angiogenesis and Altering Angiogenic Signaling

**DOI:** 10.3390/medsci12010004

**Published:** 2024-01-03

**Authors:** Maryam Ateeq, Mark Broadwin, Frank W. Sellke, M. Ruhul Abid

**Affiliations:** 1Division of Cardiothoracic Surgery, Department of Surgery, Cardiovascular Research Center, Rhode Island Hospital, The Warren Alpert Medical School of Brown University, Providence, RI 02903, USA; mateeq@alfaisal.edu (M.A.); mbroadwin@lifespan.org (M.B.); frank_sellke@brown.edu (F.W.S.); 2College of Medicine, Alfaisal University, Riyadh 11533, Saudi Arabia

**Keywords:** angiogenesis, extracellular vesicles, angiogenic signaling, exosomes, microvesicles, apoptotic bodies

## Abstract

Angiogenesis, the process of new blood vessels formation from existing vasculature, plays a vital role in development, wound healing, and various pathophysiological conditions. In recent years, extracellular vesicles (EVs) have emerged as crucial mediators in intercellular communication and have gained significant attention for their role in modulating angiogenic processes. This review explores the multifaceted role of EVs in angiogenesis and their capacity to modulate angiogenic signaling pathways. Through comprehensive analysis of a vast body of literature, this review highlights the potential of utilizing EVs as therapeutic tools to modulate angiogenesis for both physiological and pathological purposes. A good understanding of these concepts holds promise for the development of novel therapeutic interventions targeting angiogenesis-related disorders.

## 1. Introduction

The circulatory system (CS) is one of the first functioning organ systems to form in vertebrates [1]. The CS is essential to deliver oxygen and nutrients to tissues for optimal performance and functions of the tissues [2]. The process of blood vessel formation within diseased tissues has emerged as a significant enigma in the pathophysiology of cardiovascular disease. This plays an essential role during embryonic development and is now understood to have vital implications in the development of various chronic illnesses [3]. Angiogenesis, the process that involves the formation of new blood vessels from existing ones, occurs during normal growth and development of an organism, as well as in diseased conditions such as tumorigenesis, organ repair, and wound healing [4,5]. Typically, a delicate equilibrium is maintained between activation and inhibitory signals in angiogenesis. However, when this balance is disrupted, abnormal blood vessel growth occurs, leading to the development of various diseases [4].

Over the last few years, researchers have developed a keen interest in extracellular vesicles (EVs) after it was discovered that vesicles released by different cells engage in cell-to-cell communication and are capable of influencing angiogenesis based on their content and the cells they originate from [6]. In addition to traditional methods of intercellular communication, such as the secretion of autocrine or paracrine signaling factors, EVs offer an alternative mechanism for cells to communicate with neighboring and distant cells [7]. After being released from their source cells, EVs interact with the nearby microenvironment, enter the bloodstream, and travel varying distances, reach specific target sites, and influence the behavior of recipient cells by delivering bioactive cargo, thereby altering their phenotype [8]. This article provides an overview about the role of EVs from different cellular origins in angiogenesis and their impact on angiogenic signaling pathways. It also discusses the challenges and potential directions of EV therapy in angiogenesis.

## 2. Extracellular Vesicles

### 2.1. Definition and Classification

Extracellular vesicles (EVs) are membrane-bound vesicles released into the extracellular space in normal and stressed circumstances [4,9]. Mammalian cells release a diverse array of EVs that display considerable variability. EVs consist of a lipid bilayer enclosing RNA, DNA, and proteins [10]. The lipid membrane plays a crucial role in facilitating intercellular signaling by carrying ligands and receptors from the parent cells [11]. Communication through EVs appears to involve the collaborative action of their multiple components. Surface receptors and ligands are responsible for influencing the biodistribution of EVs and their ability to bind to target cells. The type of recipient cells determines the uptake of EVs by target cells. Phagocytosis is the predominant mechanism responsible for the absorption of EVs [12]. Once captured, recipient cells can integrate the protein, lipid, and genetic information conveyed by the vesicles, leading to alterations in their physiological state [9,12]. Based on the size, biogenesis, content, and function, EVs are classified into three types: exosomes, microvesicles, and apoptotic bodies (Table 1, Figure 1) [13].

### 2.2. Exosomes

#### 2.2.1. Definition, Size, and Origin

Exosomes, tiny membrane microvesicles originating from endosomes, are also known as intraluminal vesicles (ILVs) [17]. They consist of a single outer membrane and are released by all cell types, present in different body fluids [13]. Exosomes are small vesicles, measuring between 40 and 150 nm in diameter, that originate from the endosomal system [18]. The endosomal compartments undergo intraluminal budding, leading to the formation of multivesicular bodies (MVBs) that contain intraluminal vesicles (ILVs) [4,14]. Late endosomes are created through inward budding of the MVBs. During this process, specific proteins are incorporated into the invaginating membrane while the cytosolic compartments are engulfed and enclosed within the ILVs. Subsequently, the majority of ILVs are released into the extracellular space upon fusion with the plasma membrane. These released vesicles are commonly referred to as exosomes (Figure 2) [17]. Several processes contribute to the formation of exosomes, but the endothelial sorting complex required for the transport (ESCRT) pathway stands out as the central player in this biogenesis. The ESCRT system consists of four complexes: ESCRT-0, ESCRT-I, ESCRT-II, and ESCRT-III. The initiation of the pathway involves cargo delivery, determined by the ubiquitin protein checkpoint. ESCRT-0 is through phosphatidylinositol 3-phosphate PI(3)P and the recognition of ubiquitinated proteins located outside the endosomal membrane. ESCRT-I and ESCRT-II converge with ESCRT-0, playing a pivotal role in sorting cargo within the MVB and inducing bud formation. Additionally, they play regulatory functions in the formation of ESCRT-III. The ESCRT-III complex is responsible for sorting and concentrating cargo within the MVB, promoting membrane pinch-off and facilitating the release of buds into the endosome [14,19,20], while secretion is significantly influenced by SNARE proteins and their effectors such as RAB GTPases. Moreover, other mechanisms involving tetraspanin and lipids are also recognized as contributors to exosomes production [14].

#### 2.2.2. Composition/Contents

Intraluminal vesicles are small, spherical membrane-bound carriers resembling miniature versions of the parent cells. These vesicles contain a diverse array of cargo, including proteins, nucleic acids, metabolites, and other content specific to the cell of origin [17,21]. Exosomes exhibit a remarkable complexity and functional diversity, with approximately 4400 proteins, 194 lipids, 1639 mRNAs, and 764 miRNAs identified in exosomes from different cell types [17]. The key components of exosomes comprise members of the tetraspanin family (CD9, CD63, and CD81), which are involved in cell penetration, invasion, and fusion [22]. Additionally, ILVs contain an endosomal sorting complex required for transport (ESCRT) proteins Alix and TSG101, peripheral membrane protein complexes involved in exosome formation and release, as well as heat shock proteins (HSPs) involved in stress responses and antigen binding [22,23]. Other proteins such as actin, flotillins, and MVB formation proteins also play roles in exosome release and are also a part of their composition [17,22]. HSPs, CD63, ESCRT proteins, and cytoskeletal components are commonly found in all exosomes, while other proteins such as MHC Class I and II are specific to the donor cell [19]. Moreover, the exosome’s rigid bilayer membrane is composed of lipid elements, such as sphingomyelin, cholesterol, and ceramides, which influence cargo sorting, exosomes secretion, structure, and signaling [14].

Besides the selected proteins, exosomes contain a complex of nucleic acids, such as RNA, DNA, and noncoding RNA [17]. An analysis of RNA sequencing revealed that microRNAs (miRs) were the most prevalent RNA species in exosomes derived from human plasma [24]. These miRs, such as miR-214, miR29a, miR01, miR126, and miR-320, are known to play roles in essential biological processes like angiogenesis, hematopoiesis, exocytosis, and tumorigenesis. Previous studies have reported their involvement in exosomes-based cell-to-cell communication [17,25]. Additionally, other types of exosome RNA species, such as ribosomal RNA (rRNA), long noncoding RNA (lncRNA), transfer RNA (tRNA), small nuclear RNA (snRNA), and small nucleolar RNA, also influence biological processes, particularly impacting tumor development (Table 2) [14].

#### 2.2.3. Biological Function

Initially perceived as a means of cellular waste disposal, exosomes have since been recognized for their involvement in intercellular communication, cellular upkeep, and promoting the progression of tumors [13]. Furthermore, exosomes have been identified as triggers for immune responses by serving as carriers of antigens [21,40]. Within the nervous system, exosomes contribute to the survival of neurons, contributing to tissue healing and rejuvenation [41,42]. Concurrently, exosomes within the CNS have been found to harbor pathological proteins like beta amyloid peptide, superoxide dismutase, and others potentially contributing to the progression of diseases [13,43].

### 2.3. Microvesicles

#### 2.3.1. Definition, Size, and Biogenesis/Formation

Microvesicles (MVs) are vesicles enclosed by membranes, which are produced and secreted in the extracellular space when the cell’s plasma membrane undergoes outward budding or pinching and typically have a heterogeneous size ranging from 100 nm up to 1 μm [13,44]. The exact mechanisms of MV formation are not completely understood; however, it is believed to involve reorganization of the cytoskeleton and changes in symmetry of phospholipids [13]. Small GTPases are known to influence the MVs regulatory pathway and regulate contractile machinery [45]. Unlike exosome biogenesis that begins with intracellular events, MV biogenesis starts with direct outward budding of the plasma membrane. Once mature, MVs are shed from the cell’s surface through a precisely controlled pinching and scission process, ultimately releasing MVs into the extracellular space [44]. These MVs are predominantly formed in specific regions of membranes known for their lipid abundance, such as lipid rafts or caveolae domains [4]. The production of MVs is influenced by the physiological state and microenvironment of the donor cell, while the number of MVs consumed by recipient cells is determined by their own physiological state and microenvironment [46]. Moreover, the absorption of MVs requires energy, because lower temperatures inhibit the uptake of MVs [13].

#### 2.3.2. Composition and Contents

MVs predominantly contain cytosolic and plasma membrane-associated proteins, particularly those known to cluster at the plasma membrane surface, such as tetraspanin [47]. Additionally, MVs commonly include cytoskeletal proteins, heat shock proteins, integrins, and proteins with posttranslational modifications. The presence of cytosolic and plasma membrane proteins can be explained by their biogenesis process [13]. Similarly, it is expected that proteins specifically associated with different organelles such as mitochondria, the Golgi apparatus, nucleus, and endoplasmic reticulum should be less abundant in MVs, because these organelles are not involved in MV formation [48].

#### 2.3.3. Biological Function

It was believed that, like exosomes, MVs served as a method for cells to dispose of unwanted materials [9]. However, it is now recognized that MVs also participate in facilitating communication between cells, whether they are nearby or distant. The recent understanding of EVs’ biological roles has sparked greater interest among researchers to comprehend their potential for both diagnostic and therapeutic applications [13].

### 2.4. Apoptotic Bodies

#### 2.4.1. Definition, Size, and Biogenesis

Apoptotic bodies (ApoBDs), small, enclosed packets containing information and substances from dying cells, were once viewed as waste containers until they were discovered to be capable of delivering useful materials to healthy recipient cells [15]. Measuring between 500 nm and 2 µm, ApoBDs are released by dying cells. Unlike other types of EVs, apoptotic bodies are formed by the breakdown of apoptotic cells into smaller fragments [13,15]. Apoptosis, known as the natural and orderly death of cells, is tightly regulated by a cluster of genes to maintain homeostasis [49]. It is a common occurrence in various physiological processes like organ development during embryogenesis, cell differentiation, and tissue regeneration, as well as in pathophysiologic processes such as tumor formation, immune deficiency, organ atrophy, and other pathological conditions [50]. Apoptosis occurs in several stages, starting with nuclear chromatin condensation and followed by nuclear splitting, membrane blebbing, and, finally, the splitting of the cellular content into vesicles known as apoptotic bodies or, more recently, referred to as apoptosomes. The process of apoptotic cell disassembly and the removal of apoptotic material by phagocytes happen rapidly, leading to the limited in vivo presence of apoptotic bodies [15].

#### 2.4.2. Composition and Contents

The makeup of apoptotic bodies stands in stark contrast to exosomes and MVs. While exosomes and MVs lack certain components, apoptotic bodies possess a diverse array of cellular constituents: micronuclei, leftover chromatin, portion of cytosol, degraded proteins, fragments of DNA, and intact organelles but limited glycosylated proteins [13,15]. It is anticipated to contain elevated levels of nucleus-related proteins like histones, mitochondrial proteins such as HSP60 and Golgi apparatus constituents, and endoplasmic reticulum elements like GRP78 [13]. Notably, ApoBDs, unlike other microvesicles, contain substantial amounts of RNA, such as mRNAs, miRNAs, IncRNAs, and circRNAs, as well [15,51].

#### 2.4.3. Biological Function

ApoBDs play a role in enhancing the effective disposal of cellular waste through neighboring phagocytes. Additionally, ApoBDs contain bioactive substances such as microRNAs and DNA that contribute to the control of communication between cells [15].

## 3. Angiogenesis

### 3.1. Definition and Types

Angiogenesis is the process of the growth of blood vessels from the existing vasculature, allowing the delivery of oxygen and nutrients to the body’s tissues. Through this mechanism, a circulatory system is formed and remodeled into a complex vessel system that mediates a wide range of vital physiological processes, including tissue oxygenation, nutrient delivery, waste removal, immune response, temperature regulation, and the maintenance of blood pressure [52]. Two mechanisms of blood vessel formation are known: sprouting angiogenesis and intussusceptive angiogenesis [53]. 

### 3.2. Stages of Each Type

Blood vessel formation can occur through two mechanisms: sprouting angiogenesis, in which endothelial cells (EC) generate sprouts that extend towards an angiogenic signal, and intussusceptive angiogenesis, where interstitial tissue infiltrate preexisting vessels and create transvascular pillars that enlarge and divide the vessel [4,53].

Sprouting angiogenesis involves the creation of new blood vessels originating from existing capillaries [54]. The process of endothelial sprouting can be triggered by factors like low oxygen levels, injury, or the activation of angiogenic growth factors during tumor growth. One of the key molecules driving this process is vascular endothelial growth factor-A (VEGF-A). VEGF interacts with endothelial cell receptors, playing a vital role in various stages of new blood vessel formation. It has dose-dependent effects; at different concentrations, it can stimulate endothelial cell proliferation and guide their movement [55]. Beyond its role in cell growth and mobility, VEGF also acts as a potent inducer of vasodilation in existing vessels, increasing their permeability. Additionally, VEGF contributes to the breakdown of the extracellular matrix by enhancing the expression of matrix metalloproteinases and plasminogen activators, facilitating endothelial cell migration [54]. The process of sprouting angiogenesis involves multiple stages, including the enzymatic breakdown of the basement membrane, the proliferation and migration of ECs, the emergence of sprouts, the formation of branches, and the creation of tubes (Figure 3) [4].

Intussusceptive angiogenesis is another dynamic process within blood vessels that can significantly alter the architecture of the microcirculation [5]. This phenomenon was initially witnessed during the restructuring of lung capillaries after birth, where a single vessel transformed into two parallel vessels and started with the formation of slender endothelial pillars through the vessel lumen. The pillars widened and merged to form a wall through the vessel that divided the single lumen into two parallel lumens [55,56]. Intussusception is a fast process of vascular remodeling that can take place within hours or even minutes, because it is, initially, not dependent on proliferation. It has been demonstrated that pillar formation is not restricted to capillary plexuses but also occurs in smaller arteries and veins (Figure 4) [55]. Like sprouting angiogenesis, VEGF seems to play a crucial role in governing intussusceptive angiogenesis as well [57].

In a healthy tissue, the usual control of angiogenesis relies on maintaining a balance between positive and negative signaling impulses. Disruption of this equilibrium leads to the abnormal growth of blood vessels, which is a primary factor in various illnesses like cancer, atherosclerosis, corneal neovascularization, rheumatoid arthritis, and ischemic diseases [4].

### 3.3. Mechanisms Involved in Angiogenesis

Angiogenesis represents an intricate and multifaceted process that involves interactions between various cells, substances, and components of the ECM. This process comprises of four main steps, starting with (1) the breakdown of the basement membrane through proteolytic enzymes; (2) the movement of endothelial cells into the surrounding tissue and subsequent sprouting; (3) the proliferation of ECs; (4) and finally, the creation of a hollow structure, accompanied by the development of a fresh basement membrane, The recruitment of pericytes, the establishment of anastomosis, and the eventual initiation of the blood flow [58]. During the process of neovascularization, ECs undergo rearrangement of their internal support structure, displaying adhesion molecules on their surfaces such as integrins and selectins, release enzymes that degrade proteins and reshape their surrounding ECM [59].

### 3.4. Key Mediators of Angiogenesis

The factors that stimulate the growth of blood vessels include vascular endothelial growth factor (VEGF), fibroblast growth factor (FGF), tumor necrosis factor-alpha (TNF-α), transforming growth factor-beta (TGF-β), platelet-derived growth factor (PDGF), and angiopoietins (ANG). The production of these factors is done by different cell types, such as endothelial cells, fibroblasts, smooth muscle cells, platelets, inflammatory cells, and cancer cells. Additionally, the extracellular matrix (ECM) acts as a source for the sustained release of fibroblast growth factor-2 (FGF-2) [60].

Remarkably, a single growth factor, VEGF, prominently regulates the complex process of angiogenesis [61,62]. VEGF stimulates angiogenesis via several mechanisms, such as promoting the proliferation and survival of endothelial cells, enhancing the migration and invasion of these cells, and increasing chemotaxis. The clinical significance of VEGF is highlighted by its widespread production in most tumors, contributing to their growth. Furthermore, inhibiting VEGF-induced angiogenesis is emphasized for its significant impact on restraining tumor growth in vivo [63,64].

Another growth factor, FGF, is upregulated during active angiogenesis. FGF-2 and FGF-1 interact with specific receptor tyrosine kinases and promote endothelial cell migration, capillary formation, and signaling pathways mediated by protein kinase-C, phospholipase A2, and others, which further enhance the angiogenic activity of human endothelial cells [60,65,66]. Studies have shown that the interaction of FGF with FGFR-1 promotes endothelial cell migration and supports the formation of capillaries when cultured on collagen gels [67]. FGF-2, on the other hand, not only enhances the production of VEGF but also impacts the expression of the placental growth factor, showing interaction and synergy between FGF and other growth factors. FGF-2 also influences ECM remodeling during angiogenesis by affecting the synthesis of various components [60].

During angiogenesis, PDGF collaborates with other proangiogenic factors to facilitate the formation of new vessels by recruiting perivascular cells [68]. Studies have shown that, when exposed to PDGF, smooth muscle cells of the pulmonary artery upregulate the expression of miR-409-5p, which is released into EVs. These EVs are transported to pulmonary artery endothelial cells, resulting in compromised endothelial function, including a decrease in nitric oxide release. This disrupted interplay between PASMCs and PAECs, mediated by miR-409-5p, amplifies PASMC proliferation. This implies that alteration induced by PDGF in PASMCs plays a role in the advancement of vascular diseases [69].

The angiopoietins, which bind to EC-specific Tie receptors, contribute to vascular remodeling and angiogenesis; however, their exact role varies based on physiological and pathological conditions [60,70,71]. The transforming growth factor-β (TGF-β) superfamily, consisting of multiple growth factors, has both inhibitory and proangiogenic effects, depending on their interaction with other factors and cells [60]. Tumor necrosis factor-alpha (TNF-α), an inflammatory cytokine secreted mainly by activated macrophages, influences ECs directly by promoting cell differentiation and indirectly in vivo by stimulating angiogenic factor production from other cells [72].

Matrix metalloproteinases (MMPs), also known as matrixins, represent a group of enzymes that break down proteins within the ECM, playing a vital role in reshaping blood vessels, facilitating cell movement, and promoting the emergence of sprouts. The noticeable increase in MMP activity in ECs during instances of inflammation, wound healing, and tumor expansion underscore their significant contribution to both normal and abnormal angiogenesis [73,74].

The ECM has a significant impact on angiogenesis; for instance, when ECs are cultured as a single layer and need to form branching tubular structures, it has been demonstrated that this transformation requires cells to be coated with collagen, establishing an appropriate interaction with the surrounding matrix [75]. The composition and physical characteristics of the ECM, such as fiber thickness, density, and pore size, influence the formation of blood vessel sprouts [76,77,78]. Among the key cellular receptors involved in interactions with the cell matrix, integrins stand out. These transmembrane receptors are made up of two different subunits, α and β, and work as a pair to bind with ligands in the ECM. This binding enables the transmission of signals from the environment to changes in the internal cytoskeleton and various pathways for intracellular signaling. These changes affect various aspects, such as cell behaviors, movement, response to mechanical forces, and proliferation, and consequently impact a wide array of angiogenic processes [60].

## 4. Role of EVs in Angiogenesis

### 4.1. Main Sources of EVs

The promotion of blood vessel growth has been seen by EVs obtained from various cell types; however, in this study, our focus is on endothelial cells, mesenchymal stromal cells (MSCs), cardiac cells, and cancer cells [79,80]. MSCs are multipotent stem cells with the ability to develop into numerous cell types. While MSCs have been shown to enhance cardiac recovery, the long-term benefits are inconsistent. Additionally, there is a risk of tumorigenicity with MSCs [81]. However, recent research suggests that the primary benefit of mesenchymal stem cells may be attributed to their paracrine secretions, such as cytokines and extracellular vesicles [16,82]. A study has demonstrated that MSC-derived EVs increase the expression of important mediators such as vascular endothelial growth factor, fibroblast growth factor, and hepatocyte growth factor [83]. Numerous studies have shown that MSC exosomes have the potential to enhance systolic function, promote angiogenesis, and improve perfusion. Additionally, they may also decrease cardiac cell death, reduce infarct sizes, and lessen the extent of tissue scarring in animal models of heart disease [18].

Extracellular vesicles derived from cardiac progenitor cells (CPCs) have demonstrated the ability to safeguard the heart muscle from damage caused by ischemic reperfusion injury through their proangiogenic properties. The administration of CPC-EVs has proven to enhance heart function by stimulating angiogenesis in multiple preclinical animal models. This study conducted in August 2023 confirmed that CPC-EVs effectively stimulate the activation and migration of endothelial cells [84]. An essential step in the development of cancer is the growth of new blood vessels, a process known as tumor angiogenesis [85]. EVs originating from tumor cells play a significant role in influencing and sustaining favorable conditions for tumor growth. Within this context, ECs that are triggered by EVs released by cancer cells play a crucial role in promoting tumor angiogenesis [86]. Conventional drugs targeting VEGF signaling pathways are commonly employed to counteract the abnormal angiogenesis seen in various diseases. Nonetheless, these treatments have demonstrated limited effectiveness, especially in the context of cancer [85].

### 4.2. Evidence of EV Involvement in Angiogenesis

A research study conducted in 2018 examined the impact of injecting EVs into heart tissue to understand their influence on cardiac function, blood circulation, and vessel density in cases of chronic myocardial ischemia. The results indicated that injecting EVs leads to several positive outcomes: (1) enhancement of the cardiac output and stroke volume; (2) augmentation of the blood flow to areas with ischemic tissue; (3) increase in capillary and arteriolar density within the myocardium; and (4) association with elevated levels of expression of proteins such as p-MAPK/MAPK, p-eNOS/eNOS, and AKT, which will be further explained below [81]. Furthermore, two studies that investigated the impact of EVs derived from human bone mesenchymal stem cells on angiogenesis demonstrated that the application of these EVs to ischemic heart tissue resulted in improved angiogenic signaling via the Akt and ERK pathways. Moreover, these EV-treated tissues exhibited decreased levels of anti-angiogenic markers, suggesting their potential suitability for treating conditions characterized by a reduced blood flow or capillary density, such as heart failure or myocardial ischemia [87,88,89]. Another paper on glioblastoma demonstrated how vesicles play a crucial role in information exchange between malignant and vascular cells, the release of growth factors and cytokines by endothelial cells, activation of the PI3K.AKT signaling pathway, and the movement of pericytes. Consequently, these tumor-related extracellular vesicles autonomously stimulate the proliferation and migration of tumor cells [90]. Additionally, they were found to modify the tumor microenvironment to support the growth and spread of tumors, achieved through disrupting the ECM and releasing growth factors that facilitate tumor cell migration. Additionally, the vesicles contained cytokines that triggered immune and inflammatory responses, along with VEGF, which contributed to proangiogenic functions [6].

### 4.3. Modulation of Angiogenic Signaling Pathways by EVs

As previously noted, MSC-derived extracellular vesicles exhibit a strong ability to promote angiogenesis, which is linked to their release of signaling molecules in the local environment. When exposed to hypoxic conditions, the activation of nuclear factor-kB (NF-kB) and the transfer of transcriptionally active STAT3 were identified as the key mediators of angiogenesis induced by MSC exosomes [91,92]. These mediators are transferred to endothelial cells, where they enhance the translation of proteins that promote the growth of blood vessels. Exosomes carry and transfer various growth factors to the endothelial cells. Among them is Wnt, which interacts with its receptor to stimulate the transcription of molecules involved in angiogenesis. EVs also carry specific microRNAs, such as miR-31, which suppresses FIH, and miR-125, which promotes tip cell specification by decreasing Dela-like 4 (DLL4). Furthermore, platelet-derived growth factor-D (PDGF-D) and activation of the Wnt4/β-cathetin pathway were reported to play a role in exosome-mediated angiogenesis (Figure 5) [4,32,93,94].

In contrast to EVs derived from MSCs, the process initiated by endothelial cells begins with the secretion of extracellular vesicles abundant in microRNAs, such as miR214 and miR126. These molecules are transferred to the recipient ECs to trigger angiogenesis [4,95]. EVs also contain active MMPs that aid in angiogenesis by breaking down the components of the extracellular matrix [96]. Additionally, EVs hold a protein called Delta-like 4 (DLL4), which induces the internalization of Notch receptors and the formation of tip cells [97,98]. Additionally, the EVs possess β1 integrin, a surface component that initiates the Rac1-ERK1/2-ETS1 signaling cascade. This process triggers the enhanced release of CCL2, which is known to promote human endothelial cell proliferation, migration, and angiogenesis [4,99]. EVs also carry the urokinase plasminogen activator/urokinase plasminogen activator receptor (uPA/uPAR) complex, which prompts angiogenesis by generating plasmin [4]. Lastly, phosphatidylserine on the EV surface engages with CD36 and triggers Fyn kinase signaling. This, in turn, amplifies the production of intracellular oxidant radicals, inhibiting angiogenesis (Figure 6) [4,100].

Based on the cellular origin, extracellular vesicles can either have advantageous or harmful impacts on the processes of angiogenesis and tissue regeneration. Several cell types: endothelial cells, endothelial colony-forming cells, mesenchymal stem cells, platelets, and white blood cells release EVs that possess potent signals, promoting blood vessel formation [4]. However, certain EVs produced by platelets, ECs, or lymphocytes can also impede vessel growth by intensifying oxidative stress in the target cells [100,101,102].

### 4.4. Advantages and Disadvantages of EV-Mediated Angiogenesis

EVs present unique benefits in contrast to cell therapy. Unlike cell therapy, EVs do not self-replicate, which decreases the potential for tumor development and the transmission of viruses. They also offer enhanced safety through minimal immune response, extended storage duration, and convenient transport. Furthermore, EVs possess the ability to encapsulate substances, facilitating the targeted delivery of specific medications [103,104]. The presence of CD47, a “do not eat me signal”, on the exterior of EVs provides EVs an extra benefit, enabling them to acquire a tissue-healing capability and enhanced immune escape, thereby boosting their potential as vehicles for drug delivery [105]. Evidence from various fields of research has highlighted the potential of extracellular vesicles as promising therapeutic agents with significant clinical possibilities. However, the advancement of EV-based therapies is hindered by a limited understanding of precise target interactions, as well as difficulties in accurately measuring EV effectiveness and essential information regarding their distribution and pharmacokinetics. Therefore, revamping the design of EV studies is crucial to increase trust in how their effectiveness is understood, guaranteeing consistency and the ability to compare results in preclinical research [106].

## 5. Clinical Application

### 5.1. Diagnostic and Prognostic Implications of EVs in Angiogenesis-Related Diseases

Different components of EVs have been detected as potential indicators for cardiovascular diseases [106]. For instance, in diabetics, decreased levels of miRNA-126-3p and miRNA-26 within circulating EVs are linked to a higher occurrence of cardiovascular complications [107,108]. Similarly, elevated levels of EV proteins like Cystatin C, Serpin F2, and CD14 show a proportional association with an increase in the likelihood of experiencing new vascular incidents [109]. A study from 2011 indicated that the presence of CD31+/annexin V+ endothelial cell-derived EVs in the circulation is tied to a heightened risk of future major adverse cardiovascular and cerebral events in patients with stable coronary artery disease [110]. While the potential for diagnosing using EVs as biomarkers appears promising, further extensive, and comprehensive analyses are necessary to validate their reliability and practicality in clinical settings [107].

### 5.2. Therapeutic Applications of EVs in Modulating Angiogenesis

EVs display remarkable biocompatibility owing to their low toxicity and minimal immunogenic response. Interestingly, a research study demonstrated that injecting EVs led to an increased myocardial perfusion even in nonischemic myocardium. This implies that EVs administered directly to the heart could have effects beyond just the local ischemic region, potentially offering a valuable therapeutic approach [106,110]. In a study conducted in 2015, the impact of exosomes originating from rat bone marrow MSCs on cardiac function in a rat model of myocardial infarction was explored [111,112]. The study unveiled that exosomes effectively prevented cardiac cell ischemic injury by exerting their effects on the heart and blood vessels, facilitating cardiac remodeling. Furthermore, the study demonstrated the ability of MSC-secreted exosomes to mitigate ischemic–reperfusion injury of the myocardium by reducing the infarct size [113,114,115]. EVs originating from cardiac stem cells were also discovered to enhance cardiac function in an infarcted heart by reducing fibrosis, initiation angiogenesis, and stimulating the proliferation of cardiomyocytes [116].

An experiment Introduced a novel method for bone regeneration using therapeutic small extracellular vesicles (t-sEVs) loaded with therapeutic mRNAs, including VEGF-A. It was designed to enhance both angiogenic and osteogenic regeneration in critical size bone defects. The findings of the study suggested that the inclusion of VEGF-A in t-sEVs plays a significant role in stimulating angiogenesis and osteogenesis. This proposes a promising therapeutic strategy for bone regeneration and potentially other diseases, highlighting the potential of VEGF-A-loaded t-sEVs in the field of regenerative medicine [117].

Therapeutics based on EVs represent an advanced drug delivery system with several benefits. These include the capability to traverse the blood–brain barrier, improve drug pharmacokinetics, minimize the side effects related to synthetic nanocarriers, and enhance drug efficacy. Moreover, EV nanocarriers are adaptable for modifications, facilitating the incorporation of ligands to enable targeted drug delivery [118] (Table 3).

A study carried out by Xiaowei et al. investigated an additional use of EVs. Specifically, how EVs obtained from stem cells could offer protection to tissue affected by ischemic–reperfusion injuries. Ischemic tissue injury is often treated by a skin flap transplantation, which relies heavily on promoting angiogenesis to ensure flap survival. The application of SC-derived exosomes led to a significant enhancement in the survival rate of these transplanted flaps while also increasing the activity of genes that promote the formation of new blood vessels [116,120]. In relation to addressing illnesses linked to malfunctioning vascular barriers, MSC-EVs have displayed potential in lessening pulmonary vascular permeability and lung damage during hemorrhagic shock [107,121]. Similarly, in the case of cancer, exosomes released into the tumor environment can impact various aspects, such as the epithelial–mesenchymal transition (EMT), stem cell characteristics, and, most importantly, angiogenesis [122]. Given that exosome-derived noncoding RNAs (ncRNAs) and proteins play significant roles in promoting blood vessel growth in tumors, targeting these ncRNAs and proangiogenic proteins presents a potential approach for hindering the process of tumor angiogenesis [122,123]. Moreover, research has shown that ncRNAs obtained from EVs also have the potential to serve as valuable diagnostic markers. This can be attributed to the fact that they have greater stability and tissue specificity as compared to conventional protein markers [124]. Overall, multiple studies underscore the potential of extracellular vesicles as highly promising therapeutic agents for multiple diseases.

## 6. Conclusions

Angiogenesis, a critical process, holds significance for a broad spectrum of essential physiological functions, such as tissue oxygenation, nutrient distribution, waste elimination, the maintenance of blood pressure, etc. Furthermore, it plays a pivotal role in the advancement of pathological conditions like cancer. As discussed, EVs have emerged as the key agents in intercellular communication during both normal bodily functions and diseases. A growing body of research, including from our lab, highlights that EVs actively influence the process of blood vessel formation by targeting the key stages of vessel development through various mechanisms, depending on their cellular origin. Among the functional proteins found in EVs, VEGF stands out as a prominent mediator of angiogenesis. While the complexity of the EV content includes other functional proteins and miRNAs involved in angiogenesis, further research is needed to elucidate their specific roles in angiogenesis.

Although the general role of EVs in angiogenesis is well established, the precise mechanisms are not yet understood. Further investigations are required to precisely elucidate the role of each distinct EV subpopulation. A notable challenge faced by the expanding EV field is the vast diversity of vesicle subtypes and the intricate nature of studying their unique compositions and biological functions. Notably, certain EVs can either stimulate or hinder angiogenesis. To gain a better understanding of when and how EVs provide protective effects, it is essential to employ animal models to allow for the study of the specific roles of naturally occurring EVs.

Enhancing the knowledge of how EVs are formed and their diverse roles and understanding the factors that determine the distinct physical traits and functional elements transported by different EV subgroups will enable us to establish a standardized approach for preparing EVs in therapeutic contexts. Along with precise techniques for producing clinical-grade EVs and ensuring their quality, these developments have the potential to offer a more comprehensive evaluation of the balance between benefits and risks. The outcomes of ours and other labs’ ongoing studies will help elucidate the precise mechanisms by which EVs influence angiogenesis in an in vivo setting. The findings will help develop therapeutic modalities using EVs to modulate angiogenesis in pathological conditions.

## Figures and Tables

**Figure 1 medsci-12-00004-f001:**
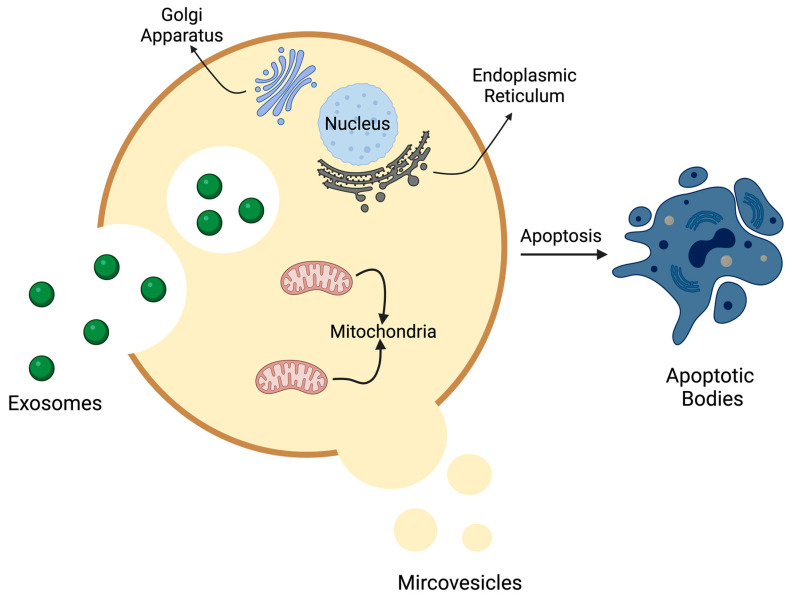
Types of extracellular vesicles: exosomes, microvesicles, and apoptotic bodies. It shows three types of extracellular vesicles. Exosomes measuring 30–150 nm are produced from multivesicular bodies via the endosomal pathway [13,14]. Microvesicles ranging from 100 nm to 1 μm are released into the extracellular space by outward budding of the plasma membrane [13,14]. Lastly, the apoptotic bodies are produced from cells undergoing programmed cell death and measure around 500 nm–2 μm [15].

**Figure 2 medsci-12-00004-f002:**
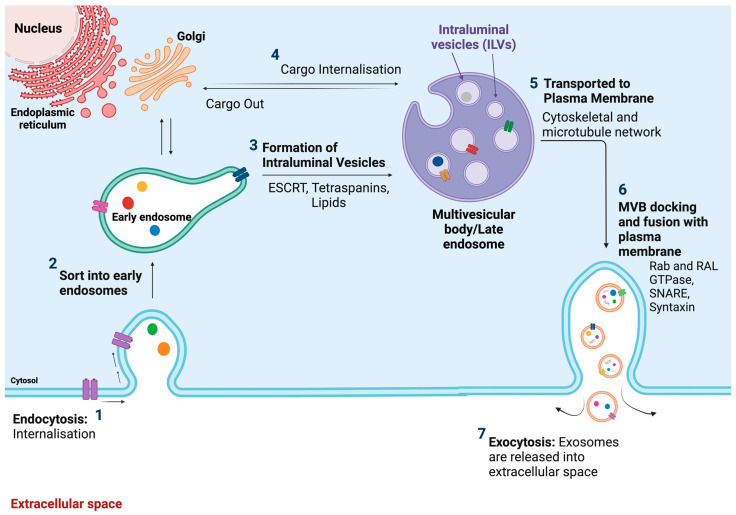
Schematic showing the step-by-step biogenesis of exosomes. Exosome production begins within the endosomal system. Early endosomes undergo maturation, eventually transforming into late endosomes or multivesicular bodies (MVBs). During this progression, the endosomal membrane invaginates, resulting in the formation of ILVs inside the organelles. The key components facilitating this process include the ESCRT machinery, tetraspanin, and lipids.

**Figure 3 medsci-12-00004-f003:**
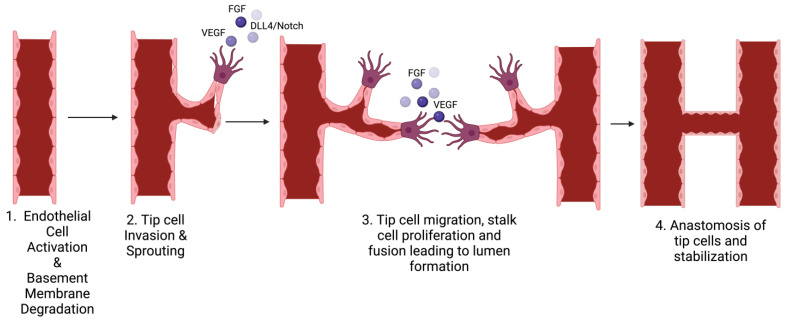
Schematic showing the stages of sprouting angiogenesis. Sprouting angiogenesis occurs when angiogenic factors trigger the activation of endothelial cells. These activated cells secrete proteases to degrade the basement membrane, generating an opening in the preexisting blood vessel, enabling the sprouting of tip cells. The sprout elongates in the direction of the stimulus. Over time, these sprouts loop together, ultimately forming a unified blood vessel.

**Figure 4 medsci-12-00004-f004:**
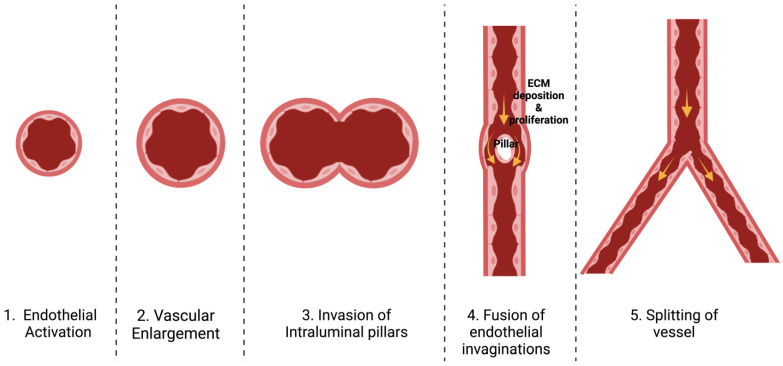
Schematic showing the stages of intussusceptive angiogenesis. Intussusceptive angiogenesis begins with the activation of the endothelial cells, leading to the expansion of the vascular network by creating pillars within the vessel lumen. This process culminates in the formation of a core where the vessel walls meet, eventually resulting in the splitting of the existing blood vessel into two.

**Figure 5 medsci-12-00004-f005:**
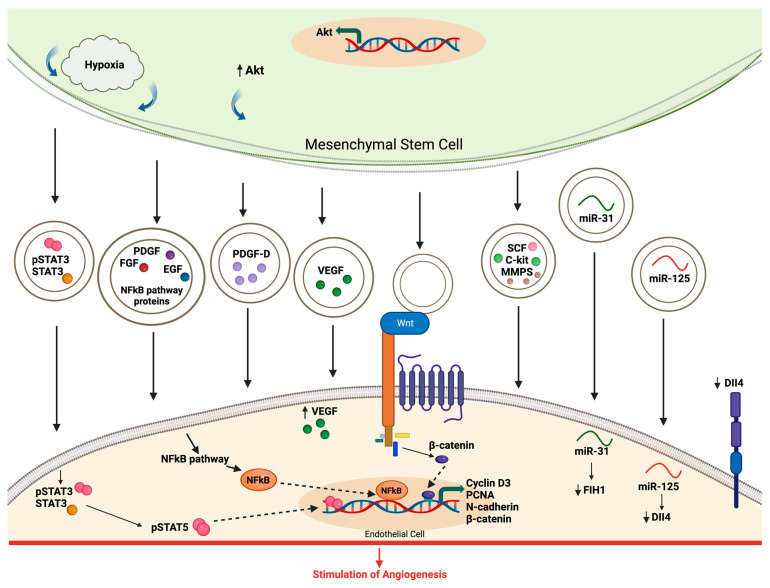
Mechanisms involved in the modulation of angiogenesis by MSC-derived EVs. Hypoxia stimulated the mesenchymal stem cells to release EVs that carry active pSTAT3 and proteins linked to the NK-kB pathway. These EVs are delivered to the EC, stimulating the production of proangiogenic proteins. EVs contain and transport various growth factors to the EC, including PDGF, FGF, EGF, VEGF, SCF, and c-kit. Wnt found in EVs, upon interaction with its receptors, encourages the transcription of molecules crucial for angiogenesis. Additionally, EVs contribute to angiogenesis by conveying various microRNAs. Notably, miR-31 operates by inhibiting the factor that suppresses HIF-1-α, while miR-125 enhances the tip cell specification by suppressing DLL4. DLL4, Delta-like 4; EGF, epidermal growth factor; FGF, fibroblast growth factor; FIH, factor inhibiting HIF-1-α; HIF, hypoxia inducible factor; NF-kB, nuclear factor-kB; PDGF, platelet-derived growth factor; SCF, stem cell factor; and VEGF, vascular endothelial growth factor.

**Figure 6 medsci-12-00004-f006:**
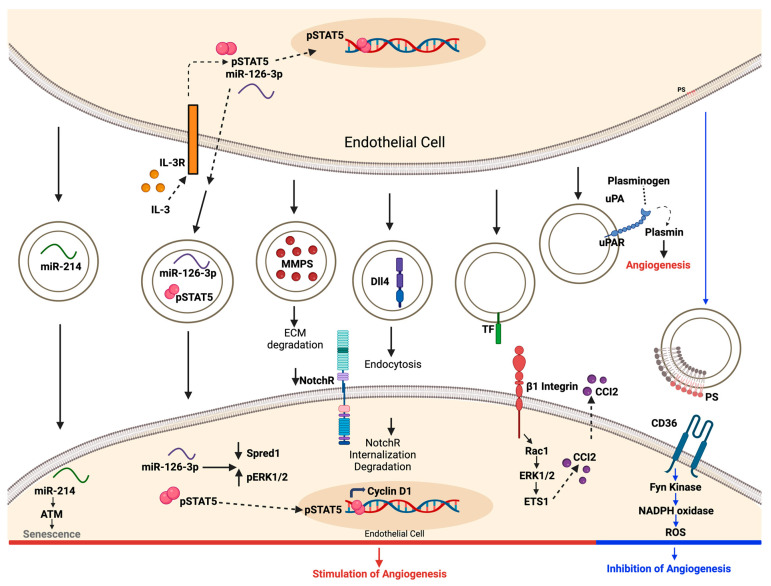
Mechanisms involved in the modulation of angiogenesis by EC-derived EVs. ECs release EVs that are enriched with miR-124 and miR126. These microRNAs are transferred to recipient ECs, where they trigger proangiogenic signaling. MMPs support this process by breaking down the components of the extracellular matrix. Additionally, the transfer of DLL4 by EVs to ECs results in Notch receptor internalization and the formation of tip cells. The tissue factor, present on the surface of EVs, interacts with β1 integrin, initiating Rac1-ERK1/2-ETS1 signaling and ultimately leading to the increased secretion of CCL2. Furthermore, these EVs transport the uPA/uPAR complex, which promotes angiogenesis by generating plasmin. Phosphatidylserine on the EV surface interacts with CD36, including Fyn kinase signaling, which enhances oxidative stress and inhibits angiogenesis. ATM, ataxia–telangiectasia-mutated; CCL2, chemokine c–c motif ligand; DLL4, Delta-like 4; ECM, extracellular matrix; ERK1/2, extracellular signal-related kinases 1 and 2; ETS1, avian erythroblastosis virus E26 homolog-1; IL-3R, interleukin-3 receptor; MMPs, matrix metalloproteinases; NotchR, Notch receptor; PS, phosphatidylserine; Rac1, Ras-related C3 botulinum toxin substrate 1; ROS, reactive oxygen species; TF, tissue factor; uPA, urokinase plasminogen activator; and uPAR, urokinase plasminogen activator receptor.

**Table 1 medsci-12-00004-t001:** Comparison of the three types of extracellular vesicles [13,15,16].

	Origin	Size	Contents	Characteristics	Biological Function
**Exosomes**	Endosomal pathway, released from Multivesicular Bodies (MVBs)	30–150 nm	miRNAs, mRNAs, proteins, lipids and nucleic acids	Small, stable and enriched in tetraspanins (CD63, CD9, CD81)	Cell- cell communication, immune modulation, transfer of genetic material & contribute to disease progression
**Microvesicles**	Plasma Membrane Budding	100 nm–1 μm	Proteins, Lipids, miRNAs, mRNAs	Heterogenous is size and content	Intercellular communication, inflammation, transmission of signals and immune response
**Apoptotic bodies**	Apoptotic cell fragmentation	500 nm–2 μm	Cellular debris, nuclear fragments, organelles, cytoplasmic components	Large and irregular in shape	Clearance of dying cells, prevent inflammation and immune regulation

**Table 2 medsci-12-00004-t002:** Overview of different types of microRNAs and IncRNAs involved in angiogenesis.

microRNA/lncRNA	Impact on Angiogenesis	Reference
Let-7	Regulate sprout formation Increases EC-mediated angiogenesis	[26]
miR-10	Enhance the ability of EC- mediated angiogenesis by suppressing anti-angiogenic genes.	[27]
miR-17-92	Facilitates cell proliferation and angiogenesis via PI3K/AKT pathway	[28]
miR-221/222	Suppress EC migration, tube formation and proliferation. Decreases EC-mediated angiogenesis	[29,30]
miR-31	Stimulate growth, migration and EC mediated angiogenesis.	[31]
miR-125	Stimulate angiogenesis by inhibiting DLL-4	[32]
miR-126	Enhance EC- mediated angiogenesis. Sustain vascular development, regeneration, and stability.	[33,34]
miR-214	Regulates angiogenesis	[35]
MALAT1	Stimulate angiogenesis	[36]
MANTIS	Induces angiogenesis	[37]
MEG3	Decrease proliferation and angiogenesis	[38]
WTAPP1	Mediate cell migration and angiogenesis	[39]

**Table 3 medsci-12-00004-t003:** Summary of recent clinical trials involving EV-based angiogenesis therapies.

Trial Title	Phase	Status	Principle Therapy	Primary Endpoints	Outcome
Antiplatelet therapy effect on extracellular vesicles in acute myocardial infarction	Phase 4	Completed	Ticagrelor and Clopidogrel	Basic science	P2Y12 antagonist ticagrelor reduces release of proinflammatory and procoagulant PEVs [119]
Effect of plasma-derived exosomes on cutaneous wound healing	Early phase 1	Unknown status	Plasma derived exosomes	Treatment	Not reported
Treatment of nonischemic cardiomyopathies by intravenous extracellular vesicles of cardiovascular progenitor cells (SECRET-HF)	Phase 1	Recruiting	Extracellular vesicle-enriched secretome of cardiovascular progenitor cells	Treatment	Not reported

## Data Availability

Not applicable.

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
