# Peer review of "Extracellular Vesicles’ Role in Angiogenesis and Altering Angiogenic Signaling"

_medsci, 2024, doi:10.3390/medsci12010004_

Round 1

Reviewer 1 Report

Comments and Suggestions for Authors

The review paper by Ateeq et al. entitled “Extracellular vesicles' role in angiogenesis and altering angiogenic signaling” is mainly devoted to a detailed description of the general characteristics of extracellular vesicles and the process of angiogenesis, which has already been described in many other review articles. On the contrary, the connections between angiogenesis and EVs are only briefly described, as well as clinical applications, and would deserve a more detailed elaboration.

Specific comments:

1.       The role of ESCRT in the biogenesis of exosomes should be explained more in detail.

2.       Via surface receptors, EVs can specifically target and be captured by recipient cells. How exactly?

3.       The last sentence of Section 2.2.2 describes various non-coding RNAs and refers to Table 1, but these RNAs are not listed in this table. Similarly, further references to Table 1 further in the text are irrelevant

4.       Section 4.3: formulation “the transcription of proteins” is not correct.

Reviewer 2 Report

Comments and Suggestions for Authors

In this review, authors summarized recent progress of EV as therapeutic tools to modulate angiogenesis and overviews roles of EVs in angiogenesis and their capacity to modulate angiogenic signaling pathways. Overall, this review is well-written and highlights recent progress of EV with interesting topics involved.  However, there are a few points that could benefit from further improvement and I would like to recommend a minor revision before acceptance.

1.     The roles of Micro-RNAs (such as the Let-7 family, miR-10, miR-17-92, miR-221/222, etc.) and LncRNAs in EV-mediated angiogenesis need a more detailed summary. It is suggested to include a table for a comprehensive overview of these small RNAs in EVs and their impact on angiogenesis.

2.     The presence and role of specific growth factor mRNAs and proteins like PDGF, VEGF, and FGF within exosomes should be expanded upon. Relevant literature and reviews on these direct angiogenic factors within exosomes should be referenced for a more thorough understanding. For example, studies on PDGF in exosomes (see doi.org/10.1515/hsz-2023-0222) and VEGF mRNA in exosomes (refer to doi.org/10.1002/advs.202302622) could be included for discussion.

3.     A summary of more recent clinical trials involving EV-based angiogenesis therapies is necessary. The author may refer to and draw inspiration from existing reviews such as doi.org/10.3390/pharmaceutics12121171, and potentially include a table summarizing recent clinical trials in this field for clarity and comprehensiveness. 

Comments on the Quality of English Language

Minor editing of English language required

Round 2

Reviewer 1 Report

Comments and Suggestions for Authors

Well done, I recommend this manuscript for publication in the journal.